**communications** engineering

# Degrees of uncertainty: conformal deep learning for non-invasive core body temperature prediction in extreme environments

Joel Strickland [1] ✉, Marco Ghisoni[2], Hannah Marshall[3,4], Thomas Whitehead[1], Bogdan Nenchev[1], Ben Pellegrini[1], Charles Phillips [1], Karl Tassenberg[1], Sarah Davey[3], Sandra Dorman[5], Joseph Sol [6], David Ferguson[7] & Gareth Conduit[1,8]

Accurate estimation of core body temperature (CBT) is essential for physiological monitoring, yet current non-invasive methods lack statistically calibrated uncertainty estimates required for safety-critical use. Here we introduce a conformal deep learning framework for real-time, non-invasive CBT prediction with calibrated uncertainty, demonstrated in high-risk heat-stress environments. Developed from over 140,000 physiological measurements across six operational domains, the model achieves a test error of 0.29 °C, outperforming the widely used ECTemp™ algorithm with a 12-fold improvement in calibrated probabilistic accuracy and statistically valid prediction intervals. Designed for integration with wearable devices, the system uses accessible physiological, demographic, and environmental inputs to support practical, confidence-informed monitoring. A customizable alert engine enables proactive safety interventions based on user-defined thresholds and model confidence. By combining deep learning with conformal prediction, this approach establishes a generalizable foundation for trustworthy, non-invasive physiological monitoring, demonstrated here for CBT under heat stress but applicable to broader safety-critical settings.

Climate change is intensifying heat events globally, driving more frequent heat-related incidents across industries[1,2]. In the United States alone, an average of 702 heat-related deaths per year were reported between 2004 and 2018[3]. Occupational settings also saw an average of 34 deaths per year between 1992 and 2022, with some years as high as 61[4]. Notably, heat-related mortality has reached new highs in recent years, coinciding with the increasing frequency and intensity of extreme heat events driven by climate change[5]. Workers in fields like construction, firefighting, military, mining, oil and gas, and warehousing are especially vulnerable due to hot environmental conditions, physically demanding tasks, and the requirement to wear heavy, impermeable protective gear[6,7]. In these thermally challenging settings, precise, continuous core body temperature (CBT) monitoring will help prevent or attenuate heat-related physical and cognitive symptoms, as well as potentially fatal conditions such as heat stroke[8] and, in the long-term, chronic kidney disease[9].

Stable CBT is essential for ensuring optimal metabolic and cell function[10]. Minor deviations from normal CBT ( + 0.5–1.0 °C) can degrade fine motor skills, selective attention, and more complex cognitive tasks such mental arithmetic and working memory[11]. Severe increases ( + 1.5–2.0 °C) are associated with heat exhaustion, moderate to severe dehydration, and heightened risk of both acute and chronic heat-related illness[12]. In extreme cases, hyperthermic states above 40 °C can result in heat stroke and multi-organ failure[13]. Given these risks and the many occupations affected, advanced, accurate, real-time CBT monitoring technologies are urgently needed to support safety and productivity.

[1]Intellegens, The Studio, Chesterton Mill, Cambridge, UK. [2]Equivital, Unit F, Buckingway Business Park, Anderson Road, Cambridge, UK. [3]Occupational and Environmental Physiology Group, Centre for Physical Activity, Sport, Exercise Sciences, Coventry University, Coventry, UK. [4]Faculty of Public Health and Policy, London School of Hygiene & Tropical Medicine, London, UK. [5]Centre for Research in Occupational Safety and Health, Laurentian University, Sudbury, Canada. [6]National Technology and Development Program, USDA Forest Service, Missoula, MT, USA. [7]Department of Kinesiology, Michigan State University, East Lansing, MI, USA. [8]Cavendish Laboratory, J.J. Thomson Avenue, Cambridge, UK. ✉e-mail: joel@intellegens.com

CBT is most accurately measured using invasive methods such as esophageal, rectal, or bladder thermometers[14]. However, these methods are not practical for continuous CBT monitoring within the workplace[15]. While gastrointestinal thermometers offer a less obtrusive alternative and demonstrate good agreement with rectal thermistors, they remain invasive as they require ingestion[16]. Non-invasive alternatives, such as tympanic infrared thermometers[17] and peripheral heat flux monitoring[18], improve practicality but lack the precision of invasive methods[19]. These limitations highlight the need for non-invasive solutions capable of delivering reliable, real-time CBT estimates, particularly in wearable formats for high-risk environments.

Achieving accurate CBT estimation from non-invasive measures is further complicated by the variability and noise inherent in physiological data. Computational methods like the Extended Kalman Filter (EKF) are commonly employed to address these challenges by integrating signals from multiple sensors[20]. However, EKF faces limitations with model inaccuracies, sensor noise, and non-Gaussian data[21]. Variants like the Unscented Kalman Filter (UKF) can enhance accuracy and convergence speed but are sensitive to dynamic changes and noise covariance estimation. This sensitivity makes their application particularly difficult during transitions between rest and activity, varying work intensities, or exposure to fluctuating and thermally challenging environments with a high risk of heat stress[22].

Building on the EKF framework, the ECTemp™ algorithm is a widely adopted solution for non-invasive CBT estimation. It models the thermoregulatory response to heat stress through sequential heart rate data alone; a practical feature for wearables[23,24]. However, ECTemp™ lacks prediction intervals with statistically guaranteed coverage, meaning it does not provide formal assurances that the true CBT value will lie within the predicted range at a specified confidence level. This is a key limitation in high-risk working environments where prediction confidence is as crucial as accuracy. ECTemp™ also assumes either a quadratic[23] or sigmoid[24] relationship between heart rate and CBT, depending on the implementation, and is primarily trained on data from young, healthy military personnel.

Although several alternative CBT estimation algorithms have been proposed[25], most similarly lack rigorous uncertainty quantification, robust generalization across diverse operational domains, or deployment-ready performance. To address these gaps, we propose a hybrid neural network combining bidirectional Long Short-Term Memory (LSTM) networks with dense layers for non-invasive CBT prediction. Optimized for wearable technology, the model integrates features commonly measured by wearables, such as heart rate, skin temperature, ambient temperature, and activity levels, along with demographic factors like age, gender, and anthropometrics. Crucially, it pairs this predictive model with conformal prediction to generate statistically calibrated prediction intervals, providing actionable confidence bounds for real-time physiological monitoring. Tested in elevated heat-risk environments, including race car driving, wildland firefighting, and mining, our model demonstrates accurate, real-time predictions across diverse and thermally challenging scenarios.

A key innovation is the use of Inductive Conformal Prediction (ICP) to provide prediction intervals with statistical guarantees for specified confidence levels. This means that, under the assumption of exchangeability in the data, the proportion of observed values falling within the prediction intervals—referred to as "coverage"—will match or exceed the desired confidence level. Our method achieves consistent uncertainty estimates by stratifying the calibration set based on predicted outputs rather than input features[26], offering reliable calibration across the full range of CBT values; a major advantage for wearable deployments in safety-sensitive applications.

Unlike traditional uncertainty estimation methods, such as Gaussian Processes, Bayesian inference, and quantile regression, which depend on specific data distribution assumptions, Conformal Prediction adapts directly to data without these constraints[27]. This flexibility allows our model to adjust prediction intervals dynamically, expanding them for uncertain predictions and narrowing them for confident ones, confirming accuracy across thermally challenging scenarios or those with a greater risk of heat stress in wearable applications.

In addition, to enhance real-time functionality, we developed a customizable alert system that uses these predictions and confidence intervals to monitor CBT continuously and alert the user to physiological thresholds to support time intervention. This system enables users to set *personalized* alert thresholds and confidence levels, adapting to the specific needs of individuals and safety protocols of different heat-risk working environments, supporting timely interventions when CBT predictions exceed defined limits, and broadening applicability across diverse industrial and occupational settings.

Our approach, combining deep learning and **S**tratification of Inductive **Co**nformal **P**rediction **E**stimates (SCOPE) provides well-calibrated prediction intervals, offering actionable insights for safety-critical applications. By achieving a test RMSE of 0.29 °C alongside reliable uncertainty quantification, our framework enables real-time, non-invasive CBT monitoring and supports proactive interventions through a customizable, confidence-driven alert system. Tailored for dynamic, high-risk working environments, this scalable solution demonstrates the potential for wearable technology in effective heat risk management across diverse occupational settings.

## Methods
### Dataset overview

The dataset consists of 140,054 measurements collected across six elevated heat-risk operational environments: wildland firefighters (WFF)[28,29], race-car drivers (RACE)[30–32], mine workers (MINE) (data collected by S. Dorman, Laurentian University), nuclear power plant workers (NUC) (data collected by H. Marshall & S. Davey, Coventry University, in collaboration with a Nuclear Power Generating Facility), explosive ordnance disposal personnel (EOD)[33] (additional EOD cooling-suit data collected by D. Dugdale-Duwell, Coventry University), and factory workers (FACT)[34]. Measurements for WFF, RACE, MINE, and NUC domains were gathered during real-world job-related tasks in field conditions, while data for EOD and FACT domains were obtained from controlled laboratory simulations replicating operational scenarios with physical activity performed both indoors and outdoors. These datasets capture a broad spectrum of climatic conditions, incorporating diverse clothing ensembles and varying metabolic activities.

All data analyzed in this study were obtained from previously approved human studies conducted in accordance with institutional ethical guidelines. Ethical approval for each dataset was granted by the respective institutional review boards, including the Coventry University Research Ethics Committee (NUC, EOD), Loughborough University Ethics Committee (FACT), Laurentian University Research Ethics Board (MINE), University of Montana–Missoula Institutional Review Board (WFF), and the Institutional Review Board at Michigan State University (RACE). All participants provided written informed consent prior to data collection, and only anonymized data were used in the analyses.

Each domain represents a unique working environment where thermal stress is a critical factor, driven by exposure to conditions that combine ambient parameters (temperature and relative humidity), clothing ensembles, and work intensity. These factors elevate the risk of heat stress, necessitating precise heart rate and core body temperature (CBT) monitoring to ensure worker safety. The domain distribution, along with subject and session counts, is summarized in Supplementary Information Fig. 1.

For each subject, measurements include physiological data (CBT, heart rate, chest skin temperature), demographic data (age, sex, mass, height, body surface area), and environmental data (ambient temperature, work intensity, clothing insulation). These data were collected using a variety of validated sensors and protocols across the contributing studies; interested readers are referred to the original publications for detailed descriptions of data acquisition methods. This broad scope ensures that the dataset captures physiological responses to a wide variety of environmental conditions, both at the group and individual levels, while also incorporating realistic variability and noise inherent in operational measurements. Such characteristics are critical for the development and deployment of models intended for real-world applications, allowing the model to account for individual variability

in thermoregulation under authentic working conditions. A full statistical summary of the variables is provided in Supplementary Information Fig. 2.

We categorize the variables into three key groups to inform model design: sequential variables (heart rate, skin temperature), which change dynamically over time; non-sequential variables (age, sex, mass, height, body surface area), which remain constant throughout a session; and semi-sequential variables (ambient temperature, work intensity, clothing insulation), which change infrequently during a session. Building on this categorization, the inclusion of work intensity, clothing insulation, and ambient temperature as semi-sequential features enables the model to directly encode the external and task-related drivers of thermal strain. Meanwhile, heart rate and skin temperature—treated as fully sequential inputs—serve as physiological proxies for internal thermoregulatory responses to exertion and the thermal environment. Importantly, heart rate reflects both activity-induced cardiovascular demand and temperature-driven thermoregulatory adjustments, meaning it can rise independently of CBT or remain elevated during rest in hot conditions. By jointly modeling these physiological signals with environmental context, the framework better disentangles activity-related and thermal contributions to CBT dynamics. This classification guides both the preprocessing steps and the architecture of the hybrid deep learning model, ensuring efficient handling of temporal data alongside static context.

### Data partitioning strategy

**Test set creation**. To evaluate the model's generalization capabilities across different domains, we constructed a test set designed to provide an unbiased assessment. All data from the Explosive Ordnance Disposal (EOD) domain, the smallest dataset with 3230 measurements, was reserved exclusively for the test set. This approach ensured that the EOD domain was completely unseen during training, validation, and calibration, making it a truly independent benchmark for evaluating model performance.

For the remaining domains—WFF, RACE, MINE, FACT, and NUC—we randomly selected four individuals and all their associated sessions from each domain to include in the test set. This ensured that the model's performance would be tested across all domains. To preserve the independence of the test set, we ensured no individual appeared in both the test set and any other subset. This resulted in a test set comprising 16,686 measurements, which accounts for 12% of the entire dataset of 140,058 measurements. The variation in the number of measurements for each domain in the test set is illustrated in Supplementary Information Fig. 3.

**Train, validation, and calibration set partitioning**. Following the definition of the test set, the remaining 123,372 measurements were partitioned into training, validation, and calibration sets with a target split ratio of 60:20:20. To maintain data integrity and prevent data leakage, we used the GroupShuffleSplit method from Scikit-learn[35], which ensures that all measurements from a single subject are allocated to the same set, maintaining the integrity of the data and supporting accurate model evaluation. This approach resulted in final proportions of 58.1%, 20.1%, and 21.8%, corresponding to 71,683, 24,749, and 26,940 measurements for the training, validation, and calibration sets, respectively. The slight deviation from the intended split ratio is due to splitting at the subject group level rather than at the individual sample level, which ensures that all related measurements remain together. The train-test comparison is shown in Supplementary Information Fig. 3, while details of the validation and calibration splits are provided in Supplementary Information Fig. 4.

Across all subsets, observed CBT values spanned a physiological range of 36.07 °C to 40.35 °C, encompassing both normal thermoregulation and heat-stress conditions. Specifically, the training data covered 36.07–40.35 °C, the validation set 36.17–39.64 °C, the calibration set 36.15–40.26 °C, and the test set 36.01–39.65 °C. Calibration and testing were therefore performed across the full temperature domain encountered in operational environments, ensuring that both uncertainty estimates and performance metrics reflect realistic use-case conditions.

### Data preprocessing

**Handling missing data**. Integrating data from various sources introduces missingness, as different variables are measured—or not measured—across datasets. Some missing data may also result from unknown or unavailable values during data collection (Supplementary Information Fig. 5 presents a missing data heatmap). To handle missing data without reduction in dataset size, we applied K-Nearest Neighbors (KNN) imputation to automatically estimate missing values. This method leverages correlations between physiological and demographic features, such as heart rate, skin temperature, and body metrics, to fill in missing data. Supplementary Information Fig. 6 shows strong correlations between key variables (e.g., heart rate, skin temperature, ambient temperature), supporting the validity of KNN imputation. This approach preserved all 140,054 data points, effectively preventing data loss.

We first normalized the training data using StandardScaler and applied KNNImputer from Scikit-learn[35] to estimate missing values based on the available data. This approach ensured that relevant feature relationships were preserved. The same scaler and imputation process were applied consistently across validation, calibration, and test sets, with transformations derived solely from the training data to prevent data leakage.

**Temporal windowing**. After splitting the data into training, validation, calibration, and test sets—and performing scaling and imputation to ensure completeness—the data was partitioned into overlapping 30-minute windows. The windows were applied at the subject-session level to maintain temporal coherence. Sessions with fewer than 30 time points or windows with missing CBT values were excluded from the analysis. Different window lengths (10, 20, 30, and 40 min) were tested, with 30 min providing the best validation performance (see Supplementary Information Table 1). After windowing, the training set contained 65,342 samples, the validation set 22,338 samples, the calibration set 23,862 samples, and the test set 13,880 samples.

### Model architecture

**LSTM-based hybrid model design**. We employed a hybrid architecture combining Long Short-Term Memory (LSTM) networks with dense layers, implemented using TensorFlow[36], to model temporal dependencies in physiological time-series data. This architecture processes both sequential and non-sequential data, capturing patterns and contextual information crucial for CBT prediction. The end-to-end data pre-processing and modelling pipeline is shown in Fig. 1.

*Sequential Data:* Heart rate and skin temperature were partitioned into 30-minute windows, created in 1-minute increments to capture fine-grained temporal dynamics, with each window consisting of historical data that was associated with the current CBT value. The windowed data was passed through a bidirectional LSTM layer to capture the temporal dynamics of physiological signals.

*Non-Sequential Data:* Demographic features (age, sex, mass, height, BSA) and environmental factors (clothing, ambient conditions, work intensity) were processed through a dense layer. Semi-sequential environmental features were treated as non-sequential due to their infrequent changes within a 30-minute window. This allowed the dense layers to reduce dimensionality while highlighting key features for CBT prediction.

By leveraging the LSTM architecture's ability to automatically learn task-relevant features, our model eliminates the need for manual feature engineering and captures complex interdependencies across physiological, demographic, and environmental variables.

Finally, the outputs from the LSTM layers (sequential data) and dense layers (non-sequential data) were concatenated into a unified feature vector. This combined vector was passed through a decision layer, which produced the final CBT prediction.

**Fig. 1 | End-to-end architecture for core body temperature (CBT) prediction, from raw data input to alert generation.** Sequential features (e.g., physiological time-series) and non-sequential features (e.g., demographic and environmental data) are first preprocessed through scaling and imputation. These processed inputs feed into a hybrid model combining Long Short-Term Memory (LSTM) and Dense layers. The model's outputs are refined by a conformal prediction layer, which provides statistically guaranteed prediction bounds at a user-defined confidence level (e.g., 95% probability that the actual CBT lies between CBT High and CBT Low). The alert layer then classifies predictions into user-defined safety states; for example, following Lousada et al.[38]: Nominal (36.5–37.75 °C, green), Advisory (36–36.5 °C or 37.75–38 °C, yellow), Caution (35–36 °C or 38–38.9 °C, orange), and Warning ( < 35 °C or > 38.9 °C, red). This modular architecture enables efficient real-time monitoring and confidence-driven alerts across diverse, safety-critical operational environments, and can be readily updated with new data or features.

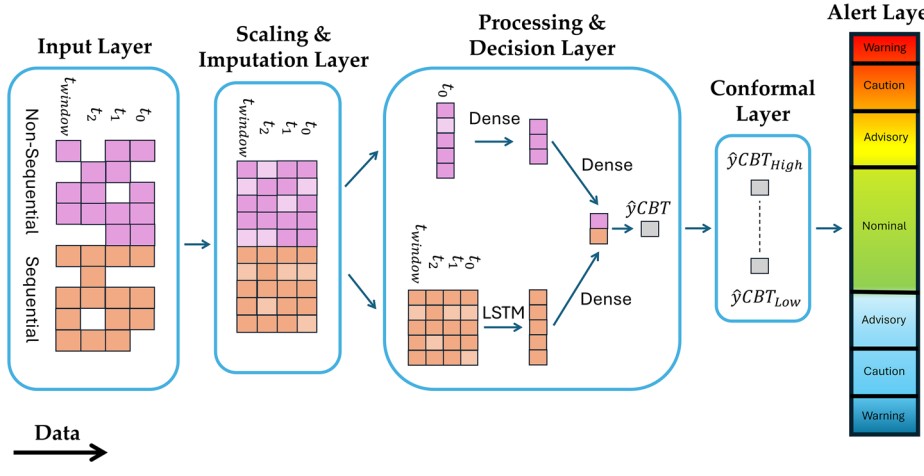

**Model training and optimization.** The model was trained using the Adam optimizer with a fixed learning rate of 0.001. Training progress was monitored through the validation set, and early stopping was applied based on validation loss, with a patience of 5 epochs. Dropout regularization (set to 0.5) was employed to mitigate overfitting. The best-performing model was found after 9 epochs, based on its performance on the validation set. Training was conducted with a batch size of 32, a maximum of 50 epochs, and 3 hidden units in the LSTM layer. All data was shuffled prior to training to ensure the model learned generalized patterns without overfitting to specific subjects or sessions.

## Uncertainty estimation and conformal prediction

**Overview of SCOPE.** SCOPE (**S**tratification of Inductive **Co**nformal **P**rediction **E**stimates) is our proposed method for estimating uncertainty in core body temperature (CBT) predictions. By dynamically stratifying the calibration set based on predicted values ($\hat{y}$), SCOPE achieves greater granularity and precision in constructing confidence intervals. This approach enables the calculation of Inductive Conformal Prediction intervals within each stratum, tailored to specific confidence levels, providing statistically robust estimates across the entire CBT range.

**Predictive-based stratification approach.** Traditional Inductive Conformal Prediction (ICP) methods do not typically involve stratification. However, Lei et al. (2018)[26] proposed stratifying calibration sets based on input features to improve the granularity of prediction intervals. SCOPE builds on this idea, but instead of stratifying by inputs, it partitions the calibration set based on ($\hat{y}$). By focusing on the prediction space, SCOPE ensures a balanced distribution of calibration points within each stratum and captures local error patterns more effectively (Fig. 2). This approach maintains the validity of confidence intervals across the full range of predicted CBT values, resulting in more accurate and reliable uncertainty estimates.

**Dynamic stratum creation.** To achieve robust uncertainty estimation—where prediction intervals consistently achieve accurate coverage across diverse conditions—SCOPE dynamically determines the number of strata required to maintain a target density of approximately 1000 points per stratum to ensure sufficient data density for reliable quantile estimates. Given a calibration set of 23,862 timepoints, the number of strata $n$

is calculated as:

$$n = \left\lceil \frac{N}{D} \right\rceil$$

where $N = 23,862$ and $D = 1000$, resulting in 24 strata.

**Nonconformity score calculation.** Within each stratum, nonconformity scores ($\epsilon_i$) are calculated as the absolute differences between the predicted ($\hat{y}^i$) and true values ($y^i$):

$$\epsilon_i = \left| \hat{y}^i - y^i \right|$$

These scores quantify how far predictions deviate from the true values. We then sort the nonconformity scores by magnitude within each stratum and compute the relevant quantiles $Q_\alpha(\epsilon_i)$ at various confidence levels (e.g., 68.27%, 95.45%, 99.73%). The quantile $Q_\alpha(\epsilon_i)$ represents the margin of error or uncertainty for the predictions in stratum $i$, based on the distribution of nonconformity scores.

**Constructing prediction intervals.** The final prediction intervals are constructed using the computed quantiles of the nonconformity scores. For a new prediction in a given stratum, the prediction interval is defined as:

$$\left[ \hat{y}^i - Q_\alpha(\epsilon_i), \; \hat{y}^i + Q_\alpha(\epsilon_i) \right]$$

This statistically guarantees, under the assumption of exchangeability between the calibration set and the test data, that the true CBT lies within the predicted interval with the specified confidence level.

**Application of SCOPE.** We applied SCOPE to a calibration set of 23,862 timepoints, dividing the prediction space into 24 strata with approximately 1000 points per bin (Fig. 2). This analysis reveals how prediction error varies across the range of predicted CBT values for three coverage levels: 68.27%, 95.45%, and 99.73%. Notably, prediction errors deviate from Gaussian assumptions and become larger at higher predicted CBT values (e.g., above 38 °C), as reflected by the increasing standard deviation in these regions.

By stratifying based on predicted values and adjusting for target density, SCOPE effectively captures these variations, producing distinct error

**Fig. 2 | Estimated prediction errors across conformal intervals for core body temperature predictions from the deep learning model, displayed for three coverage levels: 68.27% (blue), 95.45% (orange), and 99.73% (green).** Each point represents the midpoint of a temperature bin, with the y-axis showing the prediction interval bound (error margin in °C) required to achieve the specified coverage. For reference, the dashed lines represent 1σ, 2σ, and 3σ, where σ is the standard deviation of the residual errors. The prediction space was divided into 24 strata using quantile-based stratification, with approximately 1000 calibration measurements per stratum. The prediction intervals were derived from a calibration set of 23,862 measurements, capturing the variability in prediction errors across different regions of the prediction space, thereby improving the reliability of the confidence intervals.

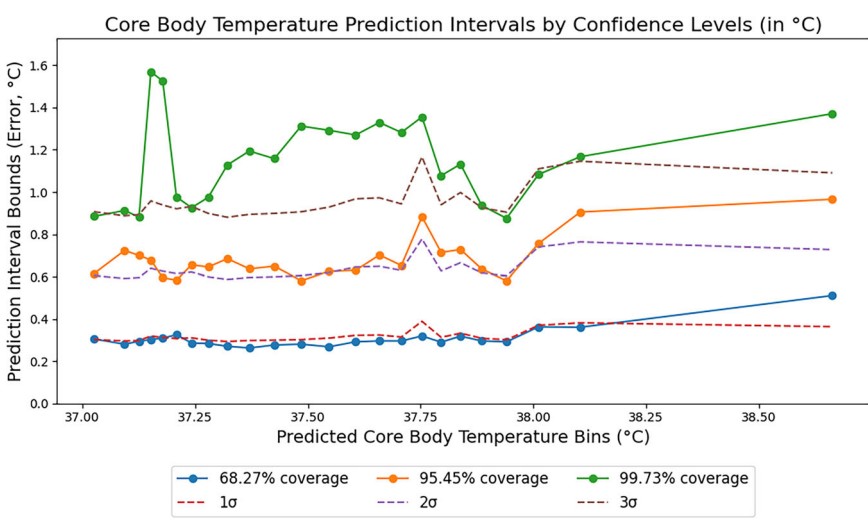

profiles across strata. Unlike methods that assume a uniform error distribution, SCOPE dynamically adapts prediction intervals to local error patterns, offering precise and reliable uncertainty estimates even in extreme temperature ranges. This balance of granularity and stability ensures consistent and valid intervals, a critical requirement for safety in high-risk environments like firefighting, race car driving, and military operations.

## Practical application of calibrated uncertainties

**SCOPE alert system.** The calibrated uncertainties provided by SCOPE can be directly integrated into a customizable alert system for continuous CBT monitoring. Each prediction includes a confidence interval with upper and lower bounds, enabling dynamic assessment against user-defined safety thresholds. For instance, a Warning alert might be triggered if the upper prediction bound exceeds 38.5 °C at a 95% confidence level.

The alert system is designed to be robust to short-term signal fluctuations. The LSTM architecture processes data over a 30 min input window, meaning sudden changes in heart rate, skin temperature, or ambient conditions have a diluted influence on the prediction. This temporal averaging reduces the impact of momentary noise and makes the model less sensitive to spurious inputs. Moreover, because the model was trained and validated on real-world data where such variability is present, the conformal prediction intervals naturally account for both model uncertainty and typical input noise. As a result, the prediction intervals—and the alert decisions based on them—already incorporate expected signal variability, helping to suppress false alarms without requiring explicit filtering or post-processing.

By integrating SCOPE's calibrated intervals, the system ensures that alerts are both precise and reliable, offering real-time, actionable insights while maintaining robustness in operational environments. This integration allows the alert system to leverage SCOPE's precise and reliable uncertainty estimates, providing actionable insights in real time. For a detailed overview of the end-to-end pipeline of the CBT alert system, refer to Fig. 1.

**Fine-tuning SCOPE.** The SCOPE method supports fine-tuning of prediction intervals to align uncertainty estimates with real-world data, optimizing coverage for practical deployment. This process uses session-specific data to examine how coverage varies across different thresholds, enabling calibration tailored to specific operational requirements.

As shown in Supplementary Information Fig. 7, session-specific coverage curves are generated to capture the relationship between confidence thresholds and observed coverage for individual subjects or domains. For a specified target coverage level (e.g., 68%), the calibration point is identified where the target intersects with the actual coverage curve. This determines

the required calibration coverage to achieve the desired confidence level. For instance, subjects such as RACER 4 and RACER 6 achieve narrower prediction intervals with required calibrated coverage values of 25% and 30%, respectively.

This approach reduces the need for large, domain-specific calibration sets by leveraging smaller subsets of session-specific data. It also takes advantage of consistent patterns in uncertainty distributions across operational contexts, ensuring robust and reliable coverage while minimizing computational overhead. Fine-tuning thus enables SCOPE to provide precise and adaptable uncertainty estimates in dynamic environments.

## Benchmarking against ECTemp™

To evaluate the performance of our model, we benchmarked it against an independent reimplementation of the ECTemp™ algorithm, a widely used method for estimating CBT from heart-rate data using an extended Kalman filter. We re-implemented the algorithm based solely on the publicly available descriptions provided by Buller et al.,[23] and Looney et al.,[24] following their published methodology and parameter settings. Specifically, CBT was initialized at 37.1 °C with an initial variance of 0.01 °C², as recommended by Buller et al.[37]. Because those authors demonstrated that the influence of the initial conditions diminishes after ~30 min, we focused our comparison on predictions made beyond this period, where initialization effects no longer affect performance.

In this implementation, the Kalman filter continuously updates the estimated CBT and its associated uncertainty as new heart-rate data are processed. We used the square root of the estimated variance (standard deviation) from the Kalman filter as the measure of prediction uncertainty, providing a real-time indication of confidence in CBT estimates.

This benchmarking was conducted solely for research comparison purposes and does not use, reproduce, or imply endorsement of any proprietary software or products from the U.S. Army Research Institute of Environmental Medicine (USARIEM) or its licensees. In accordance with editorial policy, we contacted the USARIEM communications team via the official ECTemp™ inquiry email address to inform them of the manuscript under review and the benchmarking of the ECTemp™ algorithm. Our correspondence included a summary of our methodology, findings, and an invitation to respond within five working days. No response was received. A copy of the communication was also provided to the journal editorial team.

## Statistics and reproducibility

All analyses were conducted on 140,054 timepoints from 251 subjects across six operational environments. Replicates were defined at the subject-session

level, and only sessions containing ≥10 valid timepoints were included to ensure robust estimates. Model performance was quantified using root mean squared error (RMSE) at the dataset, domain, and subject-session levels.

Pairwise model comparisons used two-sided Wilcoxon signed-rank tests to assess statistical significance of RMSE differences across subject-session replicates ($p = 0.0093$). Prediction interval calibration was evaluated using empirical coverage across multiple confidence levels, with results reported at both the subject-session and domain levels.

Data partitioning was performed using GroupShuffleSplit to prevent data leakage, ensuring all measurements from a single subject were assigned to a single subset. K-Nearest Neighbors imputation and feature scaling were fitted on the training set only and applied consistently across validation, calibration, and test sets. All models and conformal calibration procedures were evaluated exclusively on held-out data.

While full code is proprietary, the complete methodological description provided enables reproducibility in principle.

### Ethics statement

This study involved secondary analysis of previously collected, fully anonymized human data obtained from contributing institutions. All original data collection procedures were conducted under institutional ethics approvals, with informed consent obtained from all participants. The present analysis of de-identified data was conducted in accordance with institutional and journal ethical guidelines.

## Results
### Performance evaluation of models

We assessed the utility of our conformal deep learning model for real-time physiological monitoring by benchmarking it against the ECTemp™ algorithm (see Benchmarking against ECTemp™ in Methods). The evaluation spanned 140,054 measurements collected across six operational environments, representing diverse heat-stress scenarios. This dataset included 251 individuals (214 males, 36 females, and 1 unspecified), with a modal age of 35 years and modal BMI (body mass index) of 24.4. A comprehensive dataset overview, including distributions across domains, subjects, and sessions, is provided in Supplementary Information Fig. 1, with detailed statistical summaries shown in Supplementary Information Fig. 2.

Performance was quantified using root mean squared error (RMSE) across three levels: dataset (training, validation, calibration, test), domain (occupational environments), and subject-session (unique combinations of individuals and sessions). This multi-level evaluation highlights the challenges of robustness and adaptability under varying conditions.

**Dataset-level comparison.** We evaluated the predictive stability and accuracy of the deep learning model compared to ECTemp™ across four dataset partitions: training, validation, calibration, and test (see "Data partitioning strategy" in Methods). The calibration and test sets, unseen by the deep learning model during training, were reserved for independent evaluation of accuracy.

As shown in Fig. 3, the deep learning model demonstrated consistent RMSE values across all datasets, ranging from 0.29 °C to 0.31 °C. In contrast, ECTemp™ exhibited RMSE values ranging from 0.33 °C to 0.37 °C. On the calibration set, the deep learning model achieved a 17.1% lower RMSE (0.29 °C vs. 0.35 °C), and on the test set, it achieved a 14.7% lower RMSE (0.29 °C vs. 0.34 °C).

**Subject-session level comparison across domains.** We assessed predictive accuracy and consistency at the subject-session level across diverse, elevated heat-risk environments, using all available data. This granularity allowed us to evaluate how well the models handle physiological variability across individuals and sessions. RMSE values were calculated for sessions with at least 10 data points, covering occupational contexts such as wildland firefighters (WFF), race-car drivers (RACE),

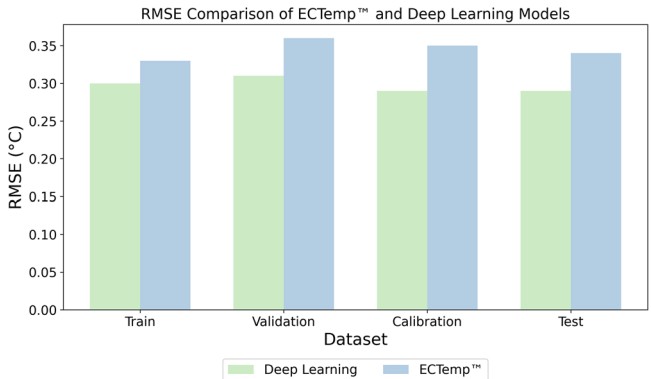

**Fig. 3 | Performance comparison between deep learning and ECTemp™ models across datasets partitions.** RMSE values for the deep learning model remain consistent across the Train, Validation, Calibration, and Test datasets (RMSE: 0.29 °C–0.31 °C), while the ECTemp™ algorithm displays greater variability (RMSE: 0.33 °C–0.37 °C). These results reflect lower prediction error and greater stability of the deep learning model across multiple data partitions.

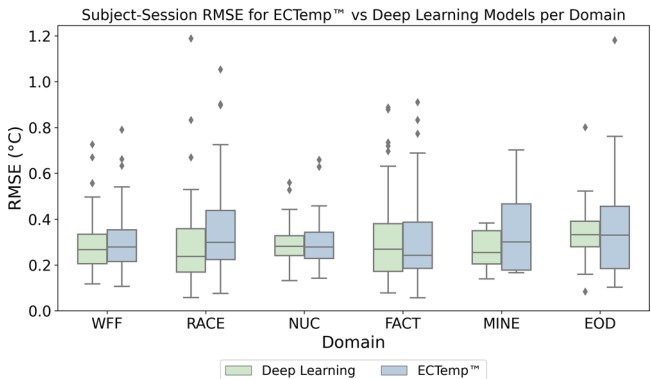

**Fig. 4 | Subject-session level RMSE comparison between Deep Learning and ECTemp™ models across domains.** Box plots display RMSE values for ECTemp™ (blue) and the Deep Learning model (green), with lower RMSE values indicating better performance. The Deep Learning model shows consistently lower and less variable RMSE across most domains, suggesting improved predictive accuracy and stability. Domains represented are wildland firefighters (WFF, $n = 152$), race-car drivers (RACE, $n = 72$), nuclear plant workers (NUC, $n = 29$), factory workers (FACT, $n = 116$), mine workers (MINE, $n = 14$), and explosive ordnance disposal technicians (EOD, $n = 31$). Only subject-sessions with at least 10 data points were included to ensure robust RMSE estimates.

nuclear plant workers (NUC), factory workers (FACT), mine workers (MINE), and explosive ordnance disposal technicians (EOD) (Fig. 4).

Box plots show that the deep learning model consistently achieved lower median RMSE values and narrower interquartile ranges (IQRs) compared to ECTemp™, indicating greater stability and predictive consistency. A Wilcoxon Signed-Rank Test confirmed a statistically significant difference in RMSE values between the two methods across subject-sessions, favoring the deep learning model ($p = 0.0093$).

**Uncertainty coverage and comparative analysis.** To evaluate uncertainty reliability, we measured the subject-session level coverage of the conformal deep learning model and ECTemp™ across various confidence thresholds on the test set (see "Uncertainty estimation and conformal prediction" in Methods). Coverage quantifies the proportion of observed values contained within the prediction intervals for a specified confidence level (e.g., 90% confidence corresponds to 90% observed coverage).

Figure 5 illustrates the test set coverage values achieved by both methods across various confidence thresholds. The conformal model

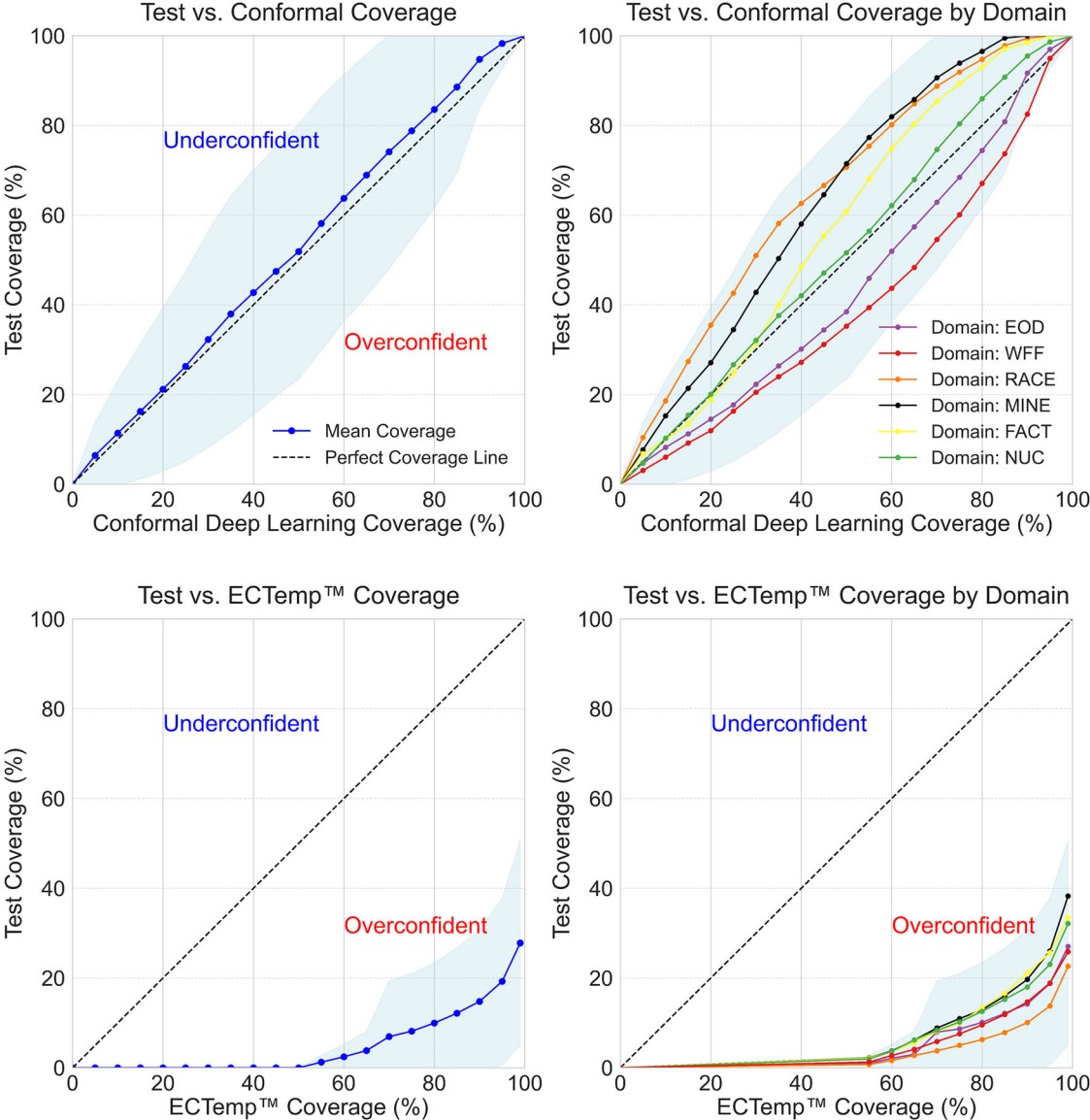

**Fig. 5 | Prediction interval coverage comparison of conformal deep learning and ECTemp™ models at the subject-session level on the test set.** The left panels show true vs. desired coverage, with the conformal model closely aligning with the perfect coverage line, indicating more reliable uncertainty estimates. The right panels illustrate domain-level coverage, where the conformal model remains consistent across datasets, while ECTemp™ shows greater variability and underestimates uncertainty. Shaded areas reflect subject-session variability. Domain key: Wildland firefighters (WFF, red), race-car drivers (RACE, orange), nuclear plant workers (NUC, green), factory workers (FACT, yellow), mine workers (MINE, black), and explosive ordnance disposal technicians (EOD, purple).

demonstrated coverage values of 11.39% at 10%, 63.76% at 60%, and 94.75% at 90%, closely aligning with the expected confidence levels. The ECTemp™ method achieved coverage values of 0% at 10%, 2.45% at 60%, and 14.78% at 90%. The area under the coverage curve (AUC) was 5247 for the conformal model, nearing perfect calibration (AUC = 5000), and representing a 12-fold improvement in probabilistic accuracy over ECTemp™ (AUC of 440). Additional details on coverage are provided in the Methods section, under 'Uncertainty estimation and conformal prediction.'

## Practical application of conformal deep learning in hot environments
### An uncertainty-driven alert framework.
The conformal deep learning model's prediction intervals provide a foundation for a real-time alert framework designed for safety-critical environments. This framework uses prediction intervals to signal alerts when predicted core body temperature (CBT) values or their associated uncertainty bounds exceed user-defined thresholds.

The framework allows users to define confidence levels and thresholds to align with specific operational needs. For instance, an alert structure informed by Lousada et al.[38] demonstrates the functionality at a 68% confidence threshold. As shown in Fig. 6, the framework classifies CBT predictions from eight subject-session pairs in the test set into four alert states—Nominal, Advisory, Caution, and Warning—based on sustained trends in predictions and uncertainty bounds.

While wider intervals often occur near critical CBT values (e.g., approaching 40 °C), this is not hardcoded; the conformal prediction method adaptively adjusts interval width wherever residual error is high, regardless of the absolute temperature (Fig. 2). Additional details on the alert system are available in the Methods section, under "SCOPE Alert System" within "Practical application of calibrated uncertainties."

### Fine-tuning uncertainty for improved performance.
The conformal deep learning model delivers well-calibrated uncertainty coverage across a diverse dataset, including elevated heat-risk conditions, variable

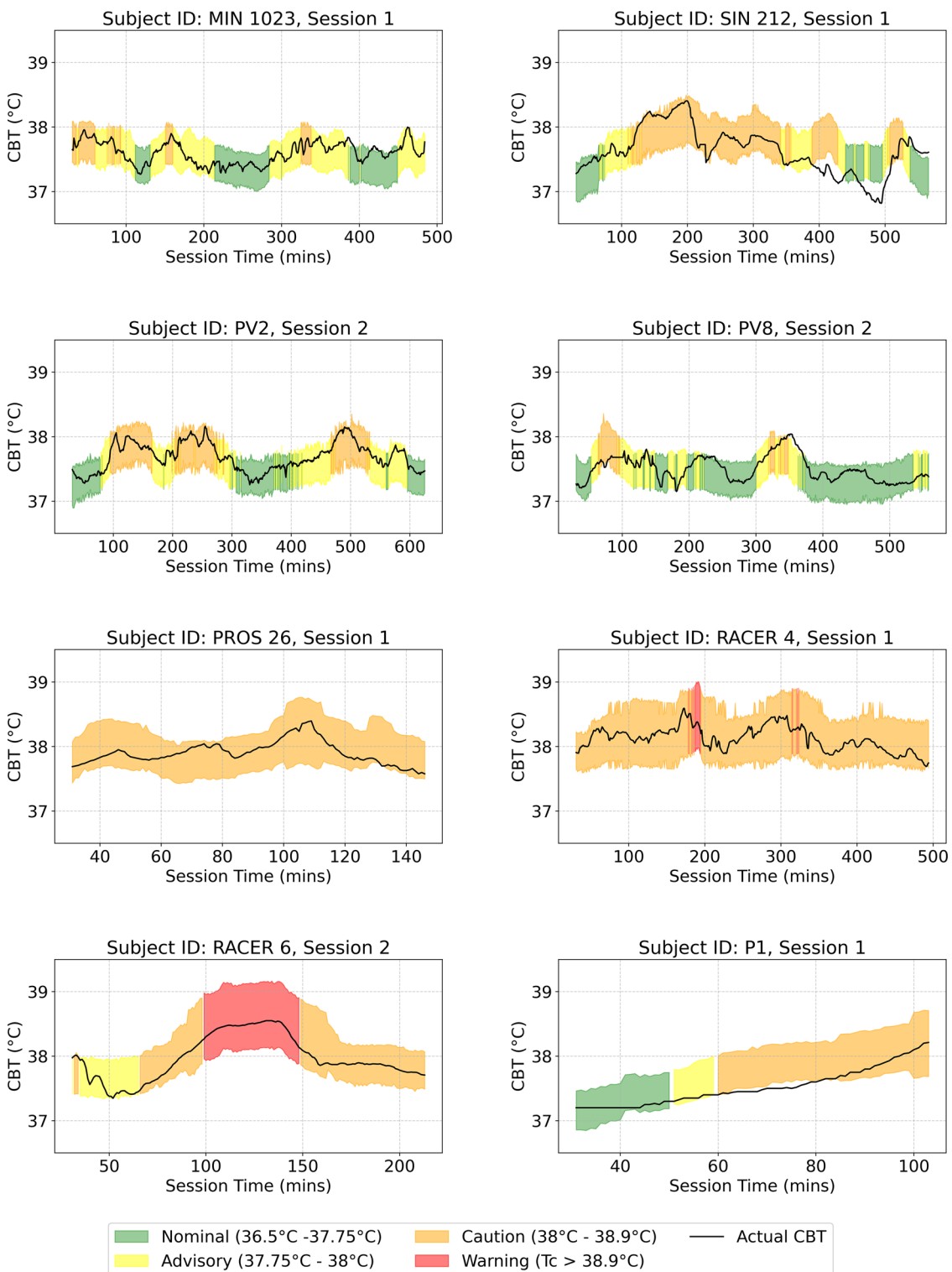

**Fig. 6 | Application of the conformal deep learning model for managing heat-related risks.** The plot shows how the model generates alerts by assessing the 68% prediction interval for core body temperature (CBT) across safety thresholds. Shaded areas represent the 68% confidence interval around each predicted CBT value, capturing the range within which the actual CBT is expected to fall. Alerts are triggered when prediction bounds exceed customizable safety thresholds, as shown in this example with unseen test subjects. Alert states are color-coded: Nominal (36.5 °C–37.75 °C, green), Advisory (37.75 °C–38 °C, yellow), Caution (38 °C–38.9 °C, orange), and Warning (above 38.9 °C, red). Confidence levels and thresholds are fully customizable, allowing the model to be adapted for various elevated heat-risk working environments where confidence-driven alerts support proactive safety interventions.

environments, and noisy data. However, subject-session level uncertainty can occasionally deviate from expected confidence levels, as shown in the shaded regions of Fig. 5. Incorporating additional domain- or individual-specific data allows prediction intervals to be locally calibrated, improving alignment with their intended confidence targets.

Global calibration, computed across all subjects and domains from the calibration set, establishes a robust baseline for generalizable uncertainty coverage (Fig. 6). In contrast, local calibration, illustrated here using subject-specific error from the test set (simulating additional field data collection), shows how a small amount of individual data can be used to recalibrate prediction intervals. This adjustment achieves true 68% coverage and better captures how well the model predicts for a given subject (Fig. 7).

This refinement narrows intervals under stable conditions, reducing unnecessary alerts, and widens them in noisier or data-sparse settings to preserve reliable coverage. These targeted adjustments maintain statistical validity while accounting for environmental and physiological variability. Additional details on the fine-tuning process are provided in the Methods section, under 'Fine-tuning SCOPE' within 'Practical application of calibrated uncertainties.'

## Discussion

This study introduces a conformal deep learning framework that advances non-invasive core body temperature (CBT) prediction in elevated heat-risk working environments. By combining bidirectional Long Short-Term Memory (LSTM) networks and dense layers with Stratification of Inductive Conformal Prediction Estimates (SCOPE), the model achieves a test RMSE of 0.29 °C, comfortably below the 0.5 °C clinical acceptance threshold referenced by Dolson et al.[25] and comparable to the 0.28 °C average RMSE they reported across validated algorithms, while outperforming ECTemp™ (0.34 °C RMSE) by 15% (Fig. 3). Crucially, unlike prior work, our model was developed using heterogeneous data spanning multiple domains, protocols, sensors, noise levels, and instances of missing input, yielding narrower interquartile ranges (IQRs) and demonstrating robust stability under realistic operational variability.

The model's generalizability was further assessed using the explosive ordnance disposal (EOD) dataset. The EOD domain, a unique fully encapsulated operational environment, was reserved entirely for testing and yielded a mean RMSE of 0.35 °C (Fig. 4), only slightly higher than the overall test RMSE. This strong out-of-domain performance demonstrates the framework's ability to extrapolate to physiologically distinct occupational settings without retraining or recalibration, underscoring its suitability for safety-critical, real-time applications. This generalizability is supported by training on data aggregated from multiple studies, each employing different validated sensors, protocols, and operational environments. By integrating these heterogeneous data sources, the model learns device- and domain-agnostic representations, enabling deployment across diverse occupational contexts and wearable systems.

These results must also be interpreted within the context of inherent uncertainty in non-invasive CBT prediction. Input features used in this study, including heart rate and chest skin temperature, as well as ground-truth CBT labels from ingestible or rectal sensors, were collected across different studies using a variety of sensors and protocols. All measurements are subject to aleatoric uncertainty arising from physiological variability and sensor noise. For example, the Equivital EQ02+ LifeMonitor demonstrates Bland–Altman accuracy limits of ±6.8bpm for heart rate and ±0.22 °C for skin temperature[39], while intestinal CBT sensors—used as ground truth in most domains—show RMSEs of 0.1–0.2 °C relative to rectal thermometry, which is regarded as the clinical gold standard[40].

This ground-truth error imposes a theoretical lower bound of ~0.1 °C RMSE, consistent with the best-case performance observed in Fig. 4. However, operational conditions introduce additional input noise from motion artifacts, environmental fluctuations, and degraded sensor quality, raising this error floor to ~0.2–0.3 °C[41,42]. Our model's test-set RMSE of 0.29 °C (Fig. 3) thus approaches this practical limit, balancing irreducible physiological variability with real-world sensor fidelity.

Beyond predictive accuracy, our framework delivers well-calibrated estimates of prediction uncertainty (Fig. 5), addressing a key gap in existing CBT algorithms[25]. Reliable uncertainty estimates build trust in model predictions and enable timely interventions, whereas poorly calibrated or overconfident estimates risk delayed responses and heat-related injuries. Uncertainty quantification is particularly crucial in scenarios characterized by high input noise, sparse data, or domain shifts (e.g., EOD), where error rates naturally increase and decision-making consequences are magnified.

To evaluate this reliability, we assessed the subject-session level coverage of the conformal deep learning model and ECTemp™ across various confidence thresholds (Fig. 5). Our model achieved an area under the coverage curve (AUC) of 5247, approaching the ideal calibration value of 5000, and demonstrated a 12-fold improvement in probabilistic accuracy over ECTemp™ (Fig. 5). These statistically guaranteed intervals not only indicate where true CBT is likely to lie but also compensate for sensor and measurement limitations, equipping decision-makers with clear, trustworthy insights into prediction reliability. Outliers observed in Fig. 4 illustrate the effects of such noise and limited representation.

Building on this uncertainty calibration, our framework provides real-time, configurable alerts aligned with operational safety protocols. Users can adjust both confidence levels and CBT thresholds, enabling a balance between sensitivity (early detection) and specificity (false alarm reduction). For example, at a 68% confidence threshold, the model correctly bounds true CBT 68% of the time, with 16% exceeding the upper limit (Fig. 6). Alerts progress stepwise (e.g., Nominal → Advisory → Caution → Warning), avoiding abrupt transitions and allowing sufficient time for intervention. This tunable approach enables users to prioritize either safety-critical sensitivity (e.g., firefighting) or operational efficiency (e.g., industrial settings), addressing the practical trade-off between early warning and minimizing unnecessary disruptions.

In practice, this tunable capability addresses a key shortcoming of traditional heat-stress management, which often relies on rigid ACGIH work–rest cycles that are designed for the "average" worker but fail to account for substantial inter- and intra-individual variation or fluctuating environmental conditions[43]. Recognizing these limitations, personalized monitoring has emerged as critical for mitigating heat-related risk, given that individuals differ widely in their thermoregulatory responses and tolerance thresholds[44]. By combining individualized CBT prediction with calibrated confidence intervals, our framework complements these existing protocols, mitigating both false positives (unnecessary restrictions) and false negatives (missed risk). This enables more targeted, adaptive interventions that enhance safety while preserving operational efficiency.

Calibration strategies further reinforce adaptability. Global calibration across all subjects and domains establishes a robust baseline for achieving generalizable coverage in diverse contexts (Fig. 6). In practice, this can be complemented by optional local calibration (Fig. 7), which incorporates a small amount of additional subject-specific field data to refine prediction intervals. This refinement narrows intervals under stable conditions to reduce false alarms, while widening them in noisier or data-sparse settings to maintain reliable coverage and prevent missed high-risk events. This balance between minimizing false alarms and ensuring timely alerts highlights the framework's ability to adapt across varied environmental and physiological conditions.

Looking forward, expanding the dataset to incorporate greater subject diversity, environmental variability, and additional physiological markers will reduce epistemic uncertainty and push RMSE toward the aleatoric limit imposed by sensor noise and natural physiological fluctuations. Future enhancements to SCOPE, such as dynamic confidence calibration that adapts coverage across different CBT ranges, could further strengthen risk management—tightening intervals in high-risk hyperthermic states where precision is most critical. Additionally, incorporating data from physiologically extreme settings, including the fully encapsulated EOD domain used exclusively for testing here, will improve model robustness. Validation in

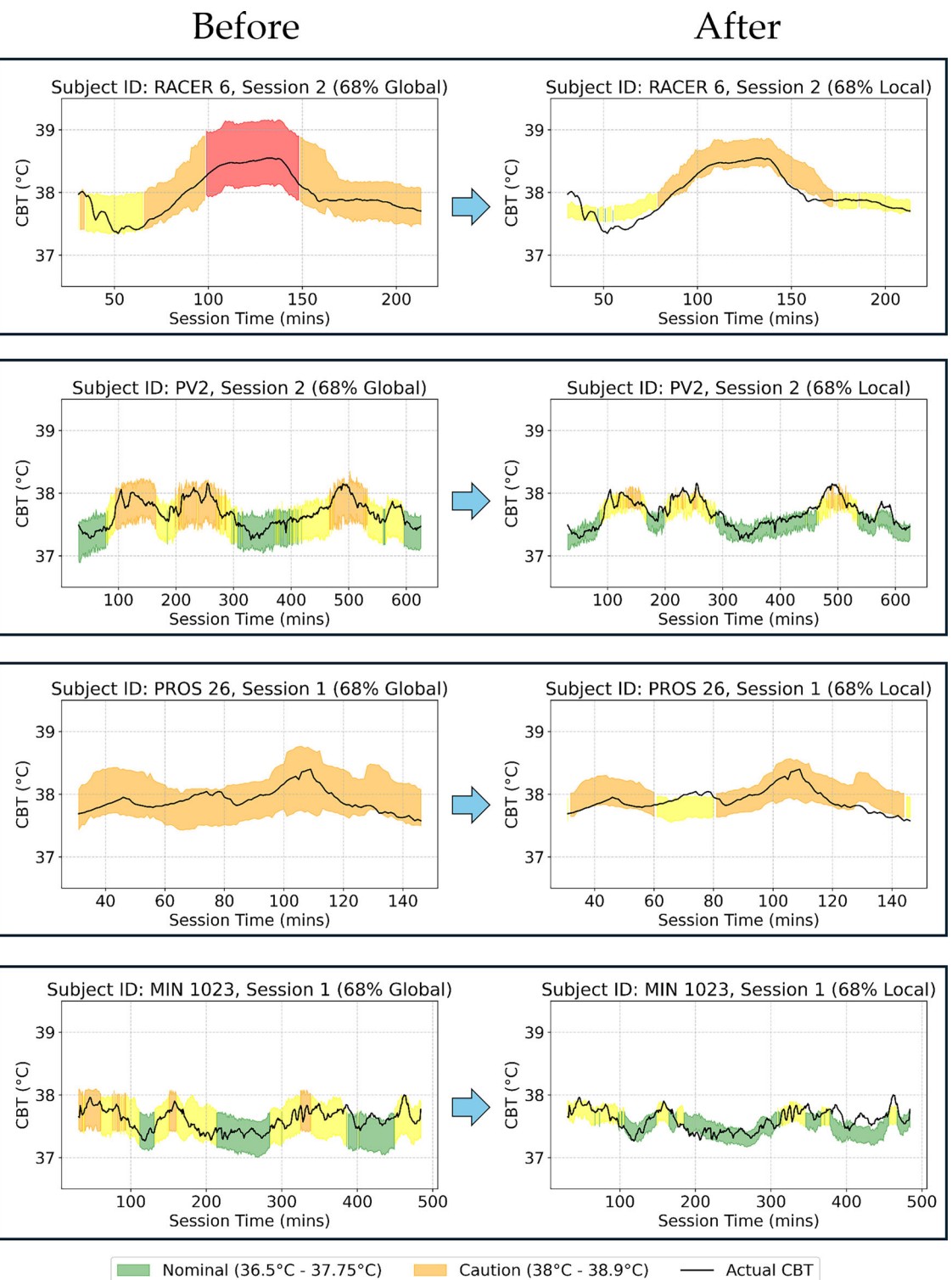

**Fig. 7 | Impact of uncertainty calibration on conformal deep learning model alerts for core body temperature (CBT).** The plot compares global calibration (all calibration data) with local calibration (subject-specific data) across four test subjects. (Left panels, 'Before') The model uses a global 68% prediction interval calibrated on all subjects and domains, resulting in wider intervals driven by baseline uncertainty. (Right panels, 'After') Prediction intervals are locally calibrated using each subject's error from the test set (simulating additional field data collection) to achieve true 68% coverage, producing individualized intervals that better capture subject-specific physiology. This refinement narrows intervals under stable conditions to reduce false alerts and widens them in noisier scenarios to maintain reliable coverage. Alerts are triggered when prediction bounds exceed user-defined thresholds, with states color-coded as Nominal (36.5 °C–37.75 °C, green), Advisory (37.75 °C–38 °C, yellow), Caution (38 °C–38.9 °C, orange), and Warning (above 38.9 °C, red).

real-world, field-deployed studies remains a priority to confirm reliability across broader operational scenarios.

Finally, while this study focused on heat stress, the framework's device-agnostic design and tolerance for missing inputs enable broader physiological monitoring applications. Extending this methodology to other temperature-related risks (e.g., hypothermia) or to indicators such as oxygen saturation, hydration status, fatigue, or respiratory rate is feasible. Because SCOPE supports recalibration for new outcomes and sensors, it provides a scalable foundation for multi-parameter, non-invasive physiological monitoring. Together, this combination of predictive accuracy, uncertainty calibration, and operational adaptability establishes a strong platform for safety-critical, real-time physiological risk management.

## Data availability

This study used six datasets representing elevated heat-risk operational environments: wildland firefighters (WFF), race-car drivers (RACE), mine workers (MINE), nuclear power plant workers (NUC), explosive ordnance disposal personnel (EOD), and factory workers (FACT). The FACT[33] dataset (PROSPIE) is available from the Loughborough University repository at https://repository.lboro.ac.uk/ndownloader/files/47227096 with https://doi.org/10.17028/rd.lboro.26076577.v1. Data from WFF[28,29], RACE[30–32], MINE (Laurentian University), NUC (Coventry University), and EOD[34] (Coventry University) are subject to third-party confidentiality or institutional restrictions and are therefore not publicly available. Requests for these datasets should be directed to the respective data owners or lead investigators cited in the manuscript, in accordance with their institutional data-sharing policies. Data are maintained by the contributing institutions and repositories listed above, in accordance with their respective data retention and security policies.

## Code availability

The code developed for this study is proprietary and cannot be shared publicly in full due to intellectual property constraints. Portions of the code, together with additional clarifications of the methodology, may be made available from the corresponding author upon reasonable request and subject to confidentiality agreements. Model development was conducted in Python 3.11 using TensorFlow v2.10.0 and Scikit-learn v1.6.1, with supporting libraries including NumPy v1.25.2, Pandas v1.5.3, SciPy v1.15.2, Matplotlib v3.10.1, Seaborn v0.13.2, and Plotly v6.1.1. The model architecture, preprocessing pipeline, hyperparameters, and training procedures are described in detail within the Methods section to support reproducibility. As the Stratification of Inductive Conformal Prediction Estimates (SCOPE) algorithm constitutes the principal methodological advance of this work, full mathematical formulations and algorithmic steps are provided in the Methods to ensure that the model and uncertainty estimation framework can be replicated and extended by others, even though the proprietary implementation cannot be distributed.

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

## Acknowledgements

The authors thank the data providers for their contributions to this study. Special acknowledgement goes to David Ferguson and Abigail Faltus for acquiring the racing car driver (RACE) dataset. The explosive ordnance disposal (EOD) data were provided by Doug Thake, derived from an MRes thesis[34] and ongoing research by Dirk Dugdale-Duwell, focusing on the efficacy of liquid cooling suits in mitigating thermal strain during explosive ordnance disposal tasks. The nuclear (NUC) data was acquired by Hannah Marshall and Sarah Davey, in collaboration with Tom Frigon at the Palo Verde Nuclear Power Generating Facility in Phoenix, Arizona, and was funded, in part, by Innovate UK under project code 13302. Sandra Dorman collected the mining (MINE) data during extensive fieldwork in operational mining environments at Laurentian University. The authors also thank Sarah Davey from Coventry University for providing the factory worker (FACT) dataset. The wildland firefighter (WFF) dataset was provided by the National Technology and Development Program, United States Department of Agriculture Forest Service. The views and conclusions contained herein are those of the authors and should not be interpreted as necessarily representing the official policies or endorsements, either expressed or implied, of the United States Department of Agriculture, Forest Service.

## Author contributions

Joel Strickland conceived the study, designed and implemented the conformal deep learning framework, developed the SCOPE algorithm, performed all machine learning experiments and analyses, and wrote the manuscript. Marco Ghisoni and Hannah Marshall were responsible for data collection and quality control. Bogdan Nenchev performed the benchmark analysis. Thomas Whitehead, Gareth Conduit, Karl Tassenberg, and Charles Phillips provided critical guidance and advice. Ben Pellegrini secured funding and offered project oversight. Sarah Davey, Sandra Dorman, David Ferguson, and Joseph Sol provided data and domain expertise to contextualize the findings. All authors reviewed the manuscript.

## Competing interests

The authors declare no competing interests.
