## [Transparent Peer Review file · Communications Engineering]

Degrees of Uncertainty: Conformal Deep Learning for Non-Invasive Core Body Temperature Prediction in Extreme Environments

Corresponding Author: Dr Joel Strickland

Version 0:

Reviewer comments:

Reviewer #1

(Remarks to the Author)

How many subjects comprise the data set?

What is the distribution of male and female?

What about BMI? There are a lot of confounding factors not mentioned in the paper.

How does the algorithm factor in the various activities mentioned in the paper that affect CBT? How sensitive is the algorithm to varied activities?

The DOI link for the PROSPIE dataset does not seem to work. Could the authors please check and revise

There are a plethora of algorithms for quantifying CBT. What makes this work unique? Can the authors please compare the RMSE of this work compared to that of published works?

Reviewer #2

(Remarks to the Author)

COMMS-25-0029-T

General comments: the authors present a very interesting analytical approach, which takes a step towards addressing complications associated with monitoring T_{core} in the field. While the analysis is interesting and promising (and very thorough), I see some distinct limitations (e.g., do we consider RMSE of 0.3°C acceptable? And the prediction intervals are still quite wide in terms of actual units [it appears they span approximately 1.0°C? which is going to yield vastly divergent physiological responses]) in the translation of this data to the field, nonetheless, it is an important step and applaud the research team for a very well presented manuscript.

My major concerns regard the underpinnings of key findings and (more so) the translation of these. I may be misinterpreting, but the model appears to have distinct requirements for data input (that being when the deep learning framework is actually utilised)? This presents as a significant limitation of the current research as would require the end-users purchase the given wearable device that the model was built upon. It appears to me that all included studies (which published data is available for) utilised the equivil physiological monitoring vest. Thus, the model assumptions are surely only valid for this given vest (under indirect assumptions that the equivil vest accurately assesses physiological strain relative to 'gold-standard' measurements). I know validity/reliability data/literature exists, but once we start compounding errors the ability of predictions to reflect reality is undoubtedly weakened? To this point, do the authors consider an RMSE of 0.3°C to be acceptable? Previous research has suggested that a change of $\geq 0.2^\circ\text{C}$ T_{core} to be of clinical relevance, and thus the inverse of this would suggest that $< 0.2^\circ\text{C}$ is clinically negligible. This is further compounded when considering 95% of the prediction interval bounds fall within $\sim 0.6^\circ\text{C}$, having the potential to result in a significant misrepresentation of the actual T_{core} observed. While this approach is clearly better than previous iterations of T_{core} prediction, it is unclear how much weight, and confidence we can actually put in to the results observed, and their ability to influence current practice. This raises interesting questions, for example, is it better to have no information and utilize current practices (e.g., Work-Rest cycles with the addition of perceptual markers of fatigue [in some cases]), or an approximate estimate of what the T_{core} response for a given individual may actually be (with the addition of requiring users to wear physiological monitoring

devices)? I am not suggesting the authors answer this question within their manuscript, but do find it of interest. Nonetheless, I imagine the value of any monitoring system is probably within the edge cases, where for whatever reason T_{core} (or some other physiological variable) is unexpectedly high (e.g., ~40°C), however, I am not sure how the model would deal with this with the current physiological data inputs?

General question: does the use of the step-wise input for alert warnings alleviate (some) issues with erroneous data triggering a 'warning alert'?

Overall, I commend the authors for a very thorough analysis of the data and an interesting piece to add to the literature-base regarding the potential of ML approaches to supersede traditional biophysical models (see Forbes et al., J Thermal Biol). At this point, it appears current research literature is approximately aligned with accuracy regarding the ability of ML models to predict T_{core}, and these models are (unsurprisingly) better than 'traditional' approaches. Again, a step-forwards, and important information, but I think there is still a long way to go before we can actually start to implement prediction modelling with confidence.

I have a few additional minor comments:

Ln 42-48: I would suggest 'softening' some of the language used within this paragraph. For example, while it can be argued that monitoring T_{core} is 'critical', T_{core} responses to occupational heat stress are not the be all and end all which dictate whether individuals suffer from heat-related illnesses (or not). Other physiological responses play significant roles in these negative outcomes and using such absolute terms may discredit their influence. I would suggest just saying 'recommended' and use a supporting reference (NIOSH 2016 should cover this).

Ln 50-52: More context is required within some of the statements used in this sentence. An increase of 0.5°C is quite different to an increase of 2.0°C and the physiological impact of these increases in T_{core}.

The Introduction is well-written and clear arguments are presented justifying this research. More of a general comment, but, interesting to conceptualise current research in consideration of industry guided work-rest cycles which are aimed at preventing T_{core} from exceeding 'safe' limits (yet heat-related injury and illness is still prevalent).

Discussion is well-written and appropriately addresses some of the limitations of the work.

Results and Methods are very thorough and I commend the authors for the work they have presented.

I recommend the manuscript for publication pending addressing some of the limitations listed above.

Version 1:

Reviewer comments:

Reviewer #2

(Remarks to the Author)

I thank the authors for addressing my concerns. I have no further comments to add.

Manuscript Title: *Degrees of Uncertainty: Conformal Deep Learning for Core Body Temperature Prediction in Extreme Environments*

Manuscript ID: COMMS-ENG-25-0029-T

We thank the Reviewers for their thoughtful and constructive feedback. We appreciate the time and care they took to assess our work, and we have revised the manuscript to address all the points raised. Below we provide a detailed response to each comment, with references to specific changes in the revised manuscript where applicable. We hope that following these revisions that the manuscript will be suitable for publication in Communications Engineering.

Reviewer #1

We thank the Reviewer for their conscientious reading of our manuscript and constructive comments. We have addressed the comments in full below, which have helped improve the manuscript. Following these changes we hope that the manuscript is now suitable for publication in Communications Engineering.

Comment 1:

How many subjects comprise the dataset? What is the distribution of male and female? What about BMI? There are a lot of confounding factors not mentioned in the paper.

Response:

There are 251 subjects in the dataset. The extended data in Figure 2 shows a suite of summary statistics, not only male/female ratio, height, age, and weight, which the Reviewer rightly points out as being important, but also a host of other whole dataset statistics. The presence of extended data Figure 2 was previously slightly opaque; we have now made this much more clear by highlighting it alongside the executive summary statistics mentioned by the Reviewer in the section “Results: Performance evaluation of models”.

Comment 2:

How does the algorithm factor in the various activities mentioned in the paper that affect CBT? How sensitive is the algorithm to varied activities?

Response:

The algorithm accounts for activity-related variability both directly and indirectly. Direct inputs include work intensity, clothing insulation, and environmental conditions are incorporated as semi-sequential input features. These capture the impact of physical activity and external stressors on CBT. Additionally, indirect inputs include heart rate and skin temperature that reflect physiological responses to activity. We have now clarified how activity is encoded in the “Methods: Dataset overview” sections.

Comment 3:

The DOI link for the PROSPIE dataset does not seem to work. Could the authors please check and revise.

Response:

We thank the Reviewer for noting this. The link has been updated to: <https://repository.lboro.ac.uk/ndownloader/files/47227096>, and its functionality was confirmed on 14 July 2025.

Comment 4:

There are a plethora of algorithms for quantifying CBT. What makes this work unique? Can the authors please compare the RMSE of this work compared to that of published works?

Response:

We thank the Reviewer for this valuable comment, which prompted us to clarify what distinguishes our approach from prior non-invasive CBT estimation algorithms. We have revised the Introduction and Discussion to emphasize these points.

In summary, our framework makes three key contributions:

- **Statistically Valid Uncertainty Estimation:** We incorporate Stratified Inductive Conformal Prediction Estimates (SCOPE) to generate prediction intervals with formal coverage guarantees. This offers actionable uncertainty estimates—essential for safety-critical applications—and is notably absent in widely used methods such as ECTemp™ (Buller et al., 2013; Looney et al., 2018) and other recent approaches (Dolson et al., 2022).
- **Generalizability Across Domains:** Our model was developed and evaluated using heterogeneous data spanning multiple operational environments (e.g., firefighting, mining, race car driving, explosive ordnance disposal), with entire domains excluded from training. This yielded robust out-of-domain performance and calibrated prediction intervals, underscoring its suitability for diverse, real-world deployment—an aspect rarely quantified in prior CBT models.
- **Comparative RMSE Performance:** As now highlighted in the Discussion, our model achieves a test RMSE of 0.29 °C, outperforming ECTemp™ (0.34 °C RMSE) by 15% and aligning closely with the 0.28 °C average RMSE reported across validated algorithms in Dolson et al. (2022). Importantly, our model achieves this while being trained on heterogeneous data incorporating multiple sensors, protocols, noise levels, and missing input, thereby enhancing operational realism and generalizability. We also note that even invasive gold-standard measurements (e.g., rectal vs. ingestible sensors) demonstrate RMSEs of ~0.1–0.2 °C (Notley et al., 2021), placing our model near the practical lower bound for non-invasive prediction.

We are grateful for the Reviewer’s suggestion, which has helped us strengthen the framing and comparative context of our contributions.

Reviewer #2

We are grateful to the Reviewer for their careful consideration of our manuscript and helpful comments. The comments have all been addressed below, and significantly improved the manuscript. Following these revisions we believe that the manuscript is now ready for publication in Communications Engineering.

Comment 1:

Do we consider RMSE of 0.3°C acceptable? Prediction intervals seem wide (~1.0°C).

Response:

Thank you for raising this important point. We agree that the acceptability of an RMSE of ~0.3°C and the width of prediction intervals must be interpreted within the context of both physiological variability and sensor limitations inherent to non-invasive core body temperature (CBT) prediction.

In response, we have revised the Discussion to clarify that our model's RMSE of 0.29°C operates near the practical error floor imposed by current sensing technologies and physiological variability. Ingestible CBT sensors—the invasive ground truth used for training—demonstrate RMSEs of 0.1–0.2°C relative to rectal thermometry, the clinical gold standard (Notley et al., 2021). This error defines a theoretical lower bound that is further elevated in operational, non-invasive settings due to input noise (e.g., motion artifacts, environmental fluctuations) and physiological variability, resulting in realistic error floors of ~0.2–0.3°C. Our model's RMSE of 0.29°C is therefore consistent with this limit and remains comfortably below the 0.5°C clinical threshold referenced by Dolson et al. (2022).

Regarding the width of the ~1.0°C prediction intervals: Extended data Fig 8 shows that ~1.0°C is ~99.73% coverage, i.e. around three standard deviations. The width corresponding to one standard deviation is therefore, as expected, comparable to the RMSE of 0.29°C. These are derived using conformal prediction methods that guarantee statistical validity under mild assumptions (e.g., exchangeability). As discussed in the manuscript, these intervals can be further fine-tuned using subject- or domain-specific calibration, which we demonstrate in Fig. 5.

Comment 2:

Model appears dependent on Equivital vest. Is generalisability limited to this device?

Response:

We thank the Reviewer for raising this important point. As clarified in the revised “Dataset overview” within the Methods section, the dataset was collected using a variety of validated sensors and protocols across the contributing studies, including but not limited to the Equivital EQ02+ LifeMonitor. Physiological inputs such as heart rate and skin temperature, as well as environmental features like ambient temperature and work intensity, were derived from multiple sources, ensuring diversity in both devices and acquisition methods.

Importantly, the SCOPE framework is inherently device-agnostic. As highlighted in the revised Discussion (paragraphs 2–3), the model was trained on data aggregated from multiple independent studies, each employing different validated sensors, protocols, and operational environments. This diversity in training data enabled the model to learn device- and domain-agnostic physiological representations.

Furthermore, the Discussion explicitly demonstrates generalizability through the explosive ordnance disposal (EOD) dataset, which was reserved entirely for testing. Despite representing a physiologically distinct, fully encapsulated operational environment unseen during training, the model achieved a mean RMSE of 0.35°C—closely matching its overall

test RMSE of 0.29 °C—showcasing its ability to extrapolate robustly to new domains without retraining or recalibration.

Finally, as noted in the Discussion (final paragraph), the framework’s device-agnostic design and tolerance for missing inputs allow it to be recalibrated or retrained for new wearable sensors and physiological measures. This adaptability is further reinforced by conformal prediction, which dynamically recalibrates uncertainty intervals when applied to novel devices or conditions.

Together, these points—explicitly addressed in the revised Discussion—demonstrate that the framework generalizes across diverse sensors and operational domains, rather than being tied to any single device such as the Equivital vest.

Comment 3:

Are model assumptions valid given sensor accuracy? Are we compounding errors?

Response:

We have now added a paragraph in the “Discussion” noting that the model’s performance summarized by the RMSE is limited by both the inherent measurement error in the invasive ingestible pill and also in the input variables. The conformal method mitigates compounded error by adjusting the uncertainty dynamically, which is discussed in the section “Uncertainty estimation and conformal prediction”.

Comment 4:

What is the practical relevance of approximate estimates vs. current strategies like work-rest cycles?

Response:

We now include in the “Discussion” that while work-rest cycles are standard practice, our system offers real-time, individualized insight that supports and enhances such heuristics. By quantifying uncertainty, our system enables more nuanced decisions (e.g., alerting earlier in edge cases or delaying unnecessarily conservative rest). This provides a layer of personalization to complement fixed-cycle policies.

Comment 5:

How does the model handle edge cases (e.g., $T_{core} \sim 40^{\circ}\text{C}$)?

Response:

We thank the Reviewer for highlighting the importance of edge-case performance, particularly in high-risk thermal ranges (e.g., $T_{core} \sim 40^{\circ}\text{C}$). As shown in Figure 4 and also the Extended Data Figure 8, our conformal prediction framework dynamically adjusts interval widths based on local prediction error. At the high end of the CBT range, where prediction error tends to increase, the model generates wider intervals. This design intentionally prioritizes safety over precision in dangerous regions, helping to reduce false negatives where underestimation could carry serious risk. We have clarified this behavior in the Results section under “Practical application of conformal deep learning in hot environments.”

Comment 6:

Does the step-wise input approach help reduce false alert triggers from noisy data?

Response:

Yes, the step-wise input approach helps mitigate false alert triggers caused by noisy data. The LSTM-based architecture captures temporal dynamics across a 30-minute window, smoothing transient fluctuations in physiological signals such as heart rate or skin temperature. In addition, alerts are triggered only when prediction intervals consistently exceed defined thresholds over time, rather than in response to isolated outliers. This persistence requirement reduces sensitivity to short-lived noise. We have clarified this mechanism in the “SCOPE Alert System” subsection of the Methods.

Minor Comments**Comment 7:**

Ln 42–48: Soften the language around ‘critical’ monitoring of Tcore.

Response:

We are grateful to the Reviewer for pointing out that the language here should be softened. Therefore, we have now changed “monitoring is critical to prevent” to now read “monitoring will help prevent”.

Comment 8:

Ln 50–52: Provide more context on physiological relevance of various increases in Tcore.

Response:

We thank the Reviewer for this suggestion. We have expanded the introduction (Lines 50–52) to clarify the physiological relevance of different levels of CBT elevation. The revised text now distinguishes between minor and moderate increases in core body temperature, linking them to specific cognitive, physical, and clinical risks. This contextualizes the importance of accurate, real-time CBT monitoring in occupational settings.

Comment 9:

Comment on industry work-rest cycles and prevalence of heat-related illness.

Response:

We thank the Reviewer for this important point. In the revised manuscript, we now explicitly discuss the limitations of current industry-standard work–rest cycles in the Discussion, highlighting their rigid, population-level assumptions and inability to account for individual variability or dynamic environmental changes. We emphasize that our framework complements these protocols by providing individualized, real-time CBT predictions with statistically calibrated uncertainty, enabling adaptive, confidence-driven interventions that address these shortcomings. Additionally, in the Introduction, we have expanded on the documented prevalence and rising incidence of heat-related illness in occupational settings, reinforcing the urgency and practical relevance of our approach for improving worker safety and mitigating heat-related risk.